# SDGM: Sparse Bayesian Classifier Based on a Discriminative Gaussian Mixture Model

## Abstract

In probabilistic classification, a discriminative model based on Gaussian mixture exhibits flexible fitting capability. Nevertheless, it is difficult to determine the number of components. We propose a sparse classifier based on a discriminative Gaussian mixture model (GMM), which is named sparse discriminative Gaussian mixture (SDGM). In the SDGM, a GMM-based discriminative model is trained by sparse Bayesian learning. This learning algorithm improves the generalization capability by obtaining a sparse solution and automatically determines the number of components by removing redundant components. The SDGM can be embedded into neural networks (NNs) such as convolutional NNs and can be trained in an end-to-end manner. Experimental results indicated that the proposed method prevented overfitting by obtaining sparsity. Furthermore, we demonstrated that the proposed method outperformed a fully connected layer with the softmax function in certain cases when it was used as the last layer of a deep NN.

## 1 Introduction

In supervised classification, probabilistic classification is an approach that assigns a class label $c$ to an input sample $\boldsymbol{x}$ by estimating the posterior probability $P(c|\boldsymbol{x})$. This approach is primarily categorized into two types of models: discriminative model and generative model. The former optimizes the posterior distribution $P(c|\boldsymbol{x})$ directly on a training set, whereas the latter finds the class conditional distribution $P(\boldsymbol{x}|c)$ and class prior $P(c)$ and subsequently derives the posterior distribution $P(c|\boldsymbol{x})$ using Bayes' rule.

The discriminative model and generative model are mutually related (Lasserre et al., 2006; Minka, 2005). According to Lasserre et al. (2006), the only difference between these models is their statistical parameter constraints. Therefore, given a certain generative model, we can derive a corresponding discriminative model. For example, the discriminative model corresponding to a unimodal Gaussian distribution is logistic regression (see Appendix A for derivation). Several discriminative models corresponding to the Gaussian mixture model (GMM) have been proposed (Axelrod et al., 2006; Bahl et al., 1996; Klautau et al., 2003; Tsai & Chang, 2002; Tsuji et al., 1999; Tüske et al., 2015; Wang, 2007). They indicate more flexible fitting capability than the generative GMM and have been applied successfully in fields such as speech recognition (Axelrod et al., 2006; Tüske et al., 2015; Wang, 2007).

The problem to address in mixture models such as the GMM is the determination of the number of components $M$. Classically, Akaike's information criterion and the Bayesian information criterion have been used; nevertheless, they require a considerable computational cost because a likelihood must be calculated for every candidate component number. In the generative GMM, methods that optimize $M$ during learning exist (Crouse et al., 2011; Štepánová & Vavrečka, 2018). However, in a discriminative GMM, a method to optimize $M$ simultaneously during learning has not been clearly formulated.

In this paper, we propose a novel GMM having two important properties: sparsity and discriminability, which is named sparse discriminative Gaussian mixture (SDGM). In the SDGM, a GMM-based discriminative model is trained by sparse Bayesian learning. This learning algorithm improves the generalization capability by obtaining a sparse solution and determines the number of components automatically by removing redundant components. Furthermore, the SDGM can be embedded into

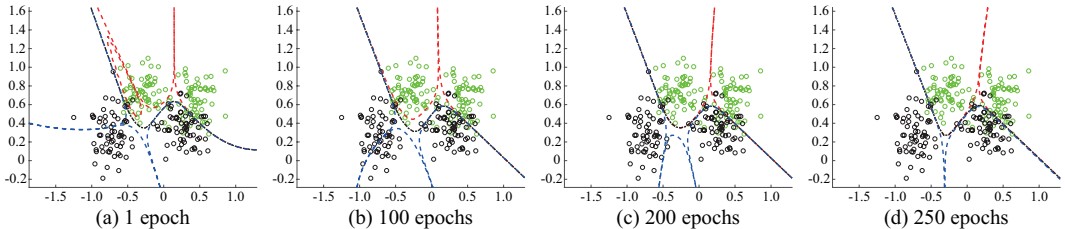

Figure 1: Snapshots of the training process of SDGM. The black and green circles are training samples from classes 1 and 2, respectively. The dashed black line is the decision boundary between classes 1 and 2 and thus satisfies $P(c = 1|\boldsymbol{x}) = (c = 2|\boldsymbol{x}) = 0.5$. The dashed blue and red lines are the boundaries between the posterior probabilities of components where $P(c, m|\boldsymbol{x}) = 0.5$.

neural networks (NNs) such as convolutional NNs and trained in an end-to-end manner with an NN. To the authors best knowledge, there is no GMM that has both of sparsity and discriminability.

The contributions of this study are as follows:

- We propose a novel sparse classifier based on a discriminative GMM. The proposed SDGM has both sparsity and discriminability, and determines the number of components automatically. The SDGM can be considered as the theoretical extension of the discriminative GMM and the relevance vector machine (RVM) (Tipping, 2001).

- This study attempts to connect both fields of probabilistic models and NNs. From the equivalence of a discriminative model based on Gaussian distribution to a fully connected layer, we demonstrate that the SDGM can be used as a module of a deep NN. We also show that the SDGM can show superior performance than the fully connected layer with a softmax function via an end-to-end learning with an NN on the image recognition task.

## 2 SPARSE DISCRIMINATIVE GAUSSIAN MIXTURE (SDGM)

An SDGM takes a continuous variable $\boldsymbol{x} \in \mathbb{R}^D$ as its input and outputs its posterior probability $P(c|\boldsymbol{x})$ for each class $c \in \{1, \ldots, C\}$. An SDGM acquires a sparse structure by removing redundant components via sparse Bayesian learning.

Figure 1 shows how the SDGM is trained by removing unnecessary components while keeping discriminability. The two-class training data are from Ripley's synthetic data (Ripley, 2006), where a Gaussian mixture model with two components is used for generating data of each class. In this training, we set the initial number of components to three for each class. As the training progresses, one of the components for each class becomes small gradually and is removed.

### 2.1 MODEL FORMULATION

The posterior probabilities for each class $c$ is calculated as follows:

$$P(c|\boldsymbol{x}) = \sum_{m=1}^{M_c} P(c, m|\boldsymbol{x}), \tag{1}$$

$$P(c, m|\boldsymbol{x}) = \frac{\pi_{cm} \exp[\boldsymbol{w}_{cm}^{\mathrm{T}}\boldsymbol{\phi}]}{\sum_{c=1}^{C} \sum_{m=1}^{M_c} \pi_{cm} \exp[\boldsymbol{w}_{cm}^{\mathrm{T}}\boldsymbol{\phi}]}, \tag{2}$$

$$\boldsymbol{\phi} = \left[1, \boldsymbol{x}^{\mathrm{T}}, x_1^2, x_1 x_2, \ldots, x_1 x_D, x_2^2, x_2 x_3, \ldots, x_D^2\right]^{\mathrm{T}}, \tag{3}$$

where $M_c$ is the number of components for class $c$ and $\pi_{cm}$ is the mixture weight that is equivalent to the prior of each component $P(c, m)$. It should be noted that we use $\boldsymbol{w}_{cm} \in \mathbb{R}^H$, which is the weight vector representing the $m$-th Gaussian component of class $c$. The dimension of $\boldsymbol{w}_{cm}$, i.e., $H$, is the same as that of $\boldsymbol{\phi}$; namely, $H = 1 + D(D + 3)/2$.

---

**Algorithm 1:** Weight updating

---

**Input:** Training data set $\mathbf{X}$ and teacher vector $\mathbf{T}$.
**Output:** Trained weight $\boldsymbol{w}$ obtained by maximizing (11).
Initialize the weights $\boldsymbol{w}$, hyperparameters $\boldsymbol{\alpha}$, mixture coefficients $\boldsymbol{\pi}$, and posterior probabilities $\boldsymbol{r}$;
**while** $\boldsymbol{\alpha}$ *have not converged* **do**
    Calculate $J$ using (9);
    **while** $\boldsymbol{r}$ *have not converged* **do**
        **while** $\boldsymbol{w}$ *have not converged* **do**
            Calculate gradients using (12);
            Calculate Hessian (13);
            Maximize (11) w.r.t. $\boldsymbol{w}$;
            Calculate $P(c, m|\boldsymbol{x}_n)$ and $P(c|\boldsymbol{x}_n)$;
        **end**
        $r_{ncm} = P(c, m|\boldsymbol{x}_n)/P(c|\boldsymbol{x}_n)$;
    **end**
    Calculate $\Lambda$ using (16);
    Update $\boldsymbol{\alpha}$ using (17);
    Update $\boldsymbol{\pi}$ using (18);
**end**

---

**Derivation.** Utilizing Gaussian distribution as a conditional distribution of $\boldsymbol{x}$ given $c$ and $m$, $P(\boldsymbol{x}|c, m)$, the posterior probability of $c$ given $\boldsymbol{x}$, $P(c|\boldsymbol{x})$, is calculated as follows:

$$P(c|\boldsymbol{x}) = \sum_{m=1}^{M_c} \frac{P(c, m)P(\boldsymbol{x}|c, m)}{\sum_{c=1}^{C} \sum_{m=1}^{M_c} P(c, m)P(\boldsymbol{x}|c, m)}, \tag{4}$$

$$P(\boldsymbol{x}|c, m) = \frac{1}{(2\pi)^{\frac{D}{2}} |\boldsymbol{\Sigma}_{cm}|^{\frac{1}{2}}} \exp\left[-\frac{1}{2}(\boldsymbol{x} - \boldsymbol{\mu}_{cm})^{\mathrm{T}} \boldsymbol{\Sigma}_{cm}^{-1}(\boldsymbol{x} - \boldsymbol{\mu}_{cm})\right], \tag{5}$$

where $\boldsymbol{\mu}_{cm} \in \mathbb{R}^D$ and $\boldsymbol{\Sigma}_{cm} \in \mathbb{R}^{D \times D}$ are the mean vector and the covariance matrix for component $m$ in class $c$. Since the calculation inside an exponential function in (5) is quadratic form, the conditional distributions can be transformed as follows:

$$P(\boldsymbol{x}|c, m) = \exp[\boldsymbol{w}_{cm}^{\mathrm{T}} \boldsymbol{\phi}], \tag{6}$$

where

$$\boldsymbol{w}_{cm} = \left[ -\frac{D}{2} \ln 2\pi - \frac{1}{2} \ln |\boldsymbol{\Sigma}_{cm}| - \frac{1}{2} \sum_{i=1}^{D} \sum_{j=1}^{D} s_{cmij} \mu_{cmi} \mu_{cmj}, \sum_{i=1}^{D} s_{cmi1} \mu_{cmi}, \cdots, \right.$$

$$\left. \sum_{i=1}^{D} s_{cmiD} \mu_{cmi}, -\frac{1}{2} s_{cm11}, -s_{cm12}, \ldots, -s_{cm1D}, -\frac{1}{2} s_{cm22}, \ldots, -\frac{1}{2} s_{cmDD} \right]^{\mathrm{T}}. \tag{7}$$

Here, $s_{cmij}$ is the $(i, j)$-th element of $\boldsymbol{\Sigma}_{cm}^{-1}$.

## 2.2 LEARNING ALGORITHM

Algorithm 1 shows the training of the SDGM. In this algorithm, the optimal weight is obtained as maximum a posteriori solution. We can obtain a sparse solution by optimizing the prior distribution set to each weight simultaneously with weight optimization.

A set of training data and target value $\{\boldsymbol{x}_n, t_{nc}\}$ $(n = 1, \cdots, N)$ is given. The target $t_{nc}$ is coded in a one-of-$K$ form, where $t_{nc} = 1$ if the $n$-th sample belongs to class $c$, $t_{nc} = 0$ otherwise. A binary random variable $z_{ncm}$ is introduced. The variable $z_{ncm} = 1$ when the $n$-th sample from class $c$ belongs to the $m$-th component. Otherwise, $z_{ncm} = 0$. This variable is required for the optimization of the mixture weight $\pi_{cm}$. We also define $\boldsymbol{\pi}$ and $\boldsymbol{z}$ as vectors that comprise $\pi_{cm}$ and $z_{ncm}$ as their elements, respectively. As the prior distribution of the weight $w_{cmh}$, we employ a

Gaussian distribution with a mean of zero. Using a different precision parameter (inverse of the variance) $\alpha_{cmh}$ for each weight $w_{cmh}$, the joint probability of all the weights is represented as follows:

$$P(\boldsymbol{w}|\boldsymbol{\alpha}) = \prod_{c=1}^{C}\prod_{m=1}^{M_c}\prod_{h=1}^{H}\sqrt{\frac{\alpha_h^{(c,m)}}{(2\pi)}}\exp\left[-\frac{1}{2}w_{cmh}{}^2\alpha_{cmh}\right], \tag{8}$$

where $\boldsymbol{w}$ and $\boldsymbol{\alpha}$ are vectors with $w_{cmh}$ and $\alpha_{cmh}$ as their elements, respectively. During learning, we update not only $\boldsymbol{w}$ but also $\boldsymbol{\alpha}$. If $\alpha_{cmh} \to \infty$, the prior (8) is 0; hence a sparse solution is obtained by optimizing $\boldsymbol{\alpha}$.

Using these variables, the expectation of the log-likelihood function over $\boldsymbol{z}$, $J$, is defined as follows:

$$J = \mathbb{E}_{\boldsymbol{z}}\left[\ln P(\mathbf{T}, \boldsymbol{z}|\mathbf{X}, \boldsymbol{w}, \boldsymbol{\pi}, \boldsymbol{\alpha})\right] = \sum_{n=1}^{N}\sum_{c=1}^{C}r_{ncm}t_{nc}\ln P(c, m|\boldsymbol{x}_n), \tag{9}$$

where $\mathbf{T}$ is a matrix with $t_{nc}$ as its element. The training data matrix $\mathbf{X}$ contains $\boldsymbol{x}_n^{\mathrm{T}}$ in the $n$-th row. The variable $r_{ncm}$ in the right-hand side corresponds to $P(m|c, \boldsymbol{x}_n)$ and can be calculated as $r_{ncm} = P(c, m|\boldsymbol{x}_n)/P(c|\boldsymbol{x}_n)$.

The posterior probability of the weight vector $\boldsymbol{w}$ is described as follows:

$$P(\boldsymbol{w}|\mathbf{T}, \boldsymbol{z}, \mathbf{X}, \boldsymbol{\pi}, \boldsymbol{\alpha}) = \frac{P(\mathbf{T}, \boldsymbol{z}|\mathbf{X}, \boldsymbol{w}, \boldsymbol{\pi}, \boldsymbol{\alpha})P(\boldsymbol{w}|\boldsymbol{\alpha})}{P(\mathbf{T}, \boldsymbol{z}|\mathbf{X}, \boldsymbol{\alpha})} \tag{10}$$

An optimal $\boldsymbol{w}$ is obtained as the point where (10) is maximized. The denominator of the right-hand side in (10) is called the evidence term, and we maximize it with respect to $\boldsymbol{\alpha}$. However, this maximization problem cannot be solved analytically; therefore we introduce the Laplace approximation described as the following procedure.

With $\boldsymbol{\alpha}$ fixed, we obtain the mode of the posterior distribution of $\boldsymbol{w}$. The solution is given by the point where the following equation is maximized:

$$\begin{aligned}\mathbb{E}_{\boldsymbol{z}}\left[\ln P(\boldsymbol{w}|\mathbf{T}, \boldsymbol{z}, \mathbf{X}, \boldsymbol{\pi}, \boldsymbol{\alpha})\right] &= \mathbb{E}_{\boldsymbol{z}}\left[\ln P(\mathbf{T}, \boldsymbol{z}|\mathbf{X}, \boldsymbol{w}, \boldsymbol{\pi}, \boldsymbol{\alpha})\right] + \ln P(\boldsymbol{w}|\boldsymbol{\alpha}) + \text{const.}\\ &= J - \boldsymbol{w}^{\mathrm{T}}\mathbf{A}\boldsymbol{w} + \text{const.}, \end{aligned} \tag{11}$$

where $\mathbf{A} = \text{diag}\,\alpha_{cmh}$. We obtain the mode of (11) via Newton's method. The gradient and Hessian required for this estimation can be calculated as follows:

$$\nabla\mathbb{E}_{\boldsymbol{z}}\left[\ln P(\boldsymbol{w}|\mathbf{T}, \boldsymbol{z}, \mathbf{X}, \boldsymbol{\pi}, \boldsymbol{\alpha})\right] = \nabla J - \mathbf{A}\boldsymbol{w}, \tag{12}$$

$$\nabla\nabla\mathbb{E}_{\boldsymbol{z}}\left[\ln P(\boldsymbol{w}|\mathbf{T}, \boldsymbol{z}, \mathbf{X}, \boldsymbol{\pi}, \boldsymbol{\alpha})\right] = \nabla\nabla J - \mathbf{A}. \tag{13}$$

Each element of $\nabla J$ and $\nabla\nabla J$ is calculated as follows:

$$\frac{\partial J}{\partial w_{cmh}} = (r_{ncm}t_{nc} - P(c, m|\boldsymbol{x}_n))\phi_h, \tag{14}$$

$$\frac{\partial^2 J}{\partial w_{cmh}\partial w_{c'm'h'}} = P(c', m'|\boldsymbol{x}_n)(P(c, m|\boldsymbol{x}_n) - \delta_{cc'mm'})\phi_h\phi_{h'}, \tag{15}$$

where $\delta_{cc'mm'}$ is a variable that takes 1 if both $c = c'$ and $m = m'$, 0 otherwise. Hence, the posterior distribution of $\boldsymbol{w}$ can be approximated by a Gaussian distribution with a mean of $\hat{\boldsymbol{w}}$ and a covariance matrix of $\Lambda$, where

$$\Lambda = -(\nabla\nabla\mathbb{E}_{\boldsymbol{z}}\left[\ln P(\hat{\boldsymbol{w}}|\mathbf{T}, \boldsymbol{z}, \mathbf{X}, \boldsymbol{\pi}, \boldsymbol{\alpha})\right])^{-1}. \tag{16}$$

Because the evidence term can be represented using the normalization term of this Gaussian distribution, we obtain the following updating rule by calculating its derivative with respect to $\alpha_{cmh}$.

$$\alpha_{cmh} \leftarrow \frac{1 - \alpha_{cmh}\lambda_{cmh}}{\hat{w}_{cmh}^2}, \tag{17}$$

where $\lambda_{cmh}$ is the diagonal component of $\Lambda$. The mixture weight $\pi_{cm}$ can be estimated using $r_{ncm}$ as follows:

$$\pi_{cm} = \frac{1}{N_c}\sum_{n=1}^{N_c}r_{ncm}, \tag{18}$$

where $N_c$ is the number of training samples belonging to class $c$. As described above, we obtain a sparse solution by alternately repeating the update of hyper-parameters, as described in (17) and (18) and the posterior distribution estimation of $\boldsymbol{w}$ using the Laplace approximation. During the procedure, the $\{c, m\}$-th component is eliminated if $\pi_{cm}$ becomes 0 or all the weights $w_{cmh}$ corresponding to the component become 0.

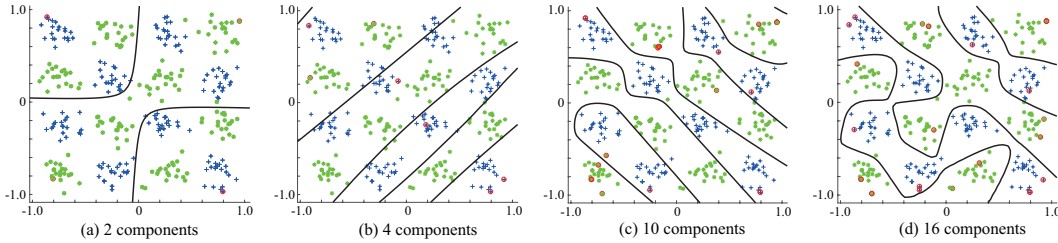

Figure 2: Changes in learned class boundaries according to number of initial components.

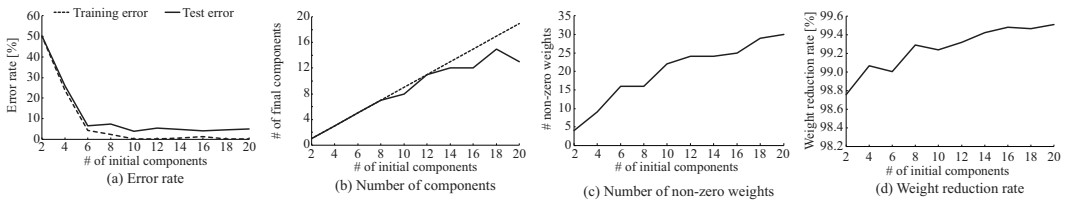

Figure 3: Evaluation results using synthetic data. (a) recognition error rate, (b) number of components after training, (c) number of nonzero weights after training, and (d) weight reduction ratio.

## 3 EXPERIMENTS

### 3.1 EVALUATION OF CHARACTERISTICS USING SYNTHETIC DATA

To evaluate the characteristics of the SDGM, we conducted classification experiments using synthetic data. The dataset comprises two classes. The data were sampled from a Gaussian mixture model with eight components for each class. The numbers of training data and test data were 320 and 1,600, respectively. The scatter plot of this dataset is shown in Figure 2.

In the evaluation, we calculated the error rates for the training data and the test data, the number of components after training, the number of nonzero weights after training, and the weight reduction ratio (the ratio of the number of the nonzero weights to the number of initial weights), by varying the number of initial components as $2, 4, 8, \ldots, 20$.

Figure 2 displays the changes in the learned class boundaries according to the number of initial components. When the number of components is small, such as that shown in Figure 2(a), the decision boundary is simple; therefore, the classification performance is insufficient. However, according to the increase in the number of components, the decision boundary fits the actual class boundaries. It is noteworthy that the SDGM learns the GMM as a discriminative model instead of a generative model; an appropriate decision boundary was obtained even if the number of components for the model is less than the actual number (e.g., 2(c)).

Figure 3 shows the evaluation results of the characteristics. Figures 3(a), (b), (c), and (d) show the recognition error rate, number of components after training, number of nonzero weights after training, and weight reduction ratio, respectively. The horizontal axis shows the number of initial components in all the graphs.

In Figure 3(a), the recognition error rates for the training data and test data are almost the same with the few number of components, and decrease according to the increase in the number of initial components while it is 2 to 6. This implied that the representation capability was insufficient when the number of components was small, and that the network could not accurately separate the classes. Meanwhile, changes in the training and test error rates were both flat when the number of initial components exceeded eight, even though the test error rates were slightly higher than the training error rate. In general, the training error decreases and the test error increases when the complexity of the classifier is increased. However, the SDGM suppresses the increase in complexity using sparse Bayesian learning, thereby preventing overfitting.

Table 1: Recognition error rate (%) and number of nonzero weights

| | Error rate (%) | | | | | Number of nonzero weights | | | | |
| | SDGM | | Baselines | | | SDGM | | Baselines | | |
| Dataset | w/ sparse | w/o sparse | LR | SVM | RVM | w/ sparse | w/o sparse | LR | SVM | RVM |
|---|---|---|---|---|---|---|---|---|---|---|
| Ripley | **9.1** | 9.9 | 11.4 | 10.6 | 9.3 | 6 | 1255 | **2** | 38 | 4 |
| Banana | **10.6** | 10.8 | 47.0 | 10.9 | 10.8 | 11.1 | 2005 | **2** | 135.2 | 11.4 |
| Waveform | 10.1 | **9.5** | 13.5 | 10.3 | 10.9 | **11.0** | 2005 | 20.73 | 146.4 | 14.6 |
| Titanic | 22.7 | 23.3 | 22.7 | **22.1** | 23.0 | 74.5 | 755 | **2.98** | 93.7 | 65.3 |
| Breast Cancer | 29.4 | 35.1 | 27.5 | **26.9** | 29.9 | 15.73 | 1005 | 8.88 | 116.7 | **6.3** |
| Normalized mean | **1.00** | 1.05 | 1.79 | 1.02 | 1.03 | 1.00 | 129.35 | **0.60** | 8.11 | 0.86 |

In Figure 3(b), the number of components after training corresponds to the number of initial components until the number of initial components is eight. When the number of initial components exceeds ten, the number of components after training tends to be reduced. In particular, eight components are reduced when the number of initial components is 20. The results above indicate the SDGM can reduce unnecessary components.

From the results in Figure 3(c), we confirm that the number of nonzero weights after training increases according to the increase in the number of initial components. This implies that the complexity of the trained model depends on the number of initial components, and that the minimum number of components is not always obtained.

Meanwhile, in Figure 3(d), the weight reduction ratio increases according to the increase in the number of initial components. This result suggests that the larger the number of initial weights, the more weights were reduced. Moreover, the weight reduction ratio is greater than 99 % in any case. The results above indicate that the SDGM can prevent overfitting by obtaining high sparsity and can reduce unnecessary components.

## 3.2 COMPARATIVE STUDY USING BENCHMARK DATA

To evaluate the capability of the SDGM quantitatively, we conducted a classification experiment using benchmark datasets. The datasets used in this experiment were Ripley's synthetic data (Ripley, 2006) (Ripley hereinafter) and four datasets cited from Rätsch et al. (2001); Banana, Waveform, Titanic, and Breast Cancer. Ripley is a synthetic dataset that is generated from a two-dimensional ($D = 2$) Gaussian mixture model, and 250 and 1,000 samples are provided for training and test, respectively. The number of classes is two ($C = 2$), and each class comprises two components. The remaining four datasets are all two-class ($C = 2$) datasets, which comprise different data size and dimensionality. Since they contain 100 training/test splits, we repeated experiments for 100 times and then calculated average statistics.

For comparison, we used three classifiers that can obtain a sparse solution: a linear logistic regression (LR) with $l_1$ constraint, a support vector machine (SVM) (Cortes & Vapnik, 1995) and a relevance vector machine (RVM) (Tipping, 2001). In the evaluation, we compared the recognition error rates for discriminability and number of nonzero weights for sparsity on the test data. The results of SVM and RVM were cited from Tipping (2001). For ablation study, we also tested our SDGM without sparse learning by omitting the update of $\boldsymbol{\alpha}$. By way of summary, the statistics were normalized by those of the SDGM and the overall mean was shown.

Table 1 shows the recognition error rates and number of nonzero weights for each method. The results in Table 1 show that the SDGM achieved an equivalent or greater accuracy compared with the SVM and RVM on average. The SDGM is developed based a Gaussian mixture model and is particularly effective for data where a Gaussian distribution can be assumed, such as the Ripley dataset. On the number of nonzero weights, understandably, the LR showed the smallest number since it is a linear model. Among the remaining nonlinear classifiers, the SDGM achieved relatively small number of nonzero weights thanks to its sparse Bayesian learning. The results above indicated that the SDGM demonstrated generalization capability and a sparsity simultaneously.

## 3.3 IMAGE CLASSIFICATION

In this experiment, the SDGM is embedded into a deep neural network. Since the SDGM is differentiable with respect to the weights, SDGM can be embedded into a deep NN as a module and is

Table 2: Recognition error rates (%) on image classification

|          | MNIST ($D = 2$) | MNIST ($D = 10$) | Fashion-MNIST | CIFAR-10 |
|----------|-----------------|------------------|---------------|----------|
| Softmax  | 3.19            | 1.01             | 8.78          | 11.07    |
| SDGM     | **2.43**        | **0.72**         | **8.30**      | **10.05** |

trained in an end-to-end manner. In particular, the SDGM plays the same role as the softmax function since the SDGM calculates the posterior probability of each class given an input vector. We can show that a fully connected layer with the softmax is equivalent to the discriminative model based on a single Gaussian distribution for each class by applying a simple transformation (see Appendix A), whereas the SDGM is based on the Gaussian mixture model.

To verify the difference between them, we conducted image classification experiments. Using a CNN with a softmax function as a baseline, we evaluated the capability of SDGM by replacing softmax with the SDGM.

### 3.3.1 DATASETS AND EXPERIMENTAL SETUPS

We used the following datasets and experimental settings in this experiment.

**MNIST**: This dataset includes 10 classes of handwritten binary digit images of size $28 \times 28$ (LeCun et al., 1998). We used 60,000 images as training data and 10,000 images as testing data. As a feature extractor, we used a simple CNN that consists of five convolutional layers with four max pooling layers between them and a fully connected layer. To visualize the learned CNN features, we first set the output dimension of the fully connected layer of the baseline CNN as two ($D = 2$). Furthermore, we tested by increasing the output dimension of the fully connected layer from two to ten ($D = 10$).

**Fashion-MNIST**: Fashion-MNIST (Xiao et al., 2017) includes 10 classes of binary fashion images with a size of $28 \times 28$. It includes 60,000 images for training data and 10,000 images for testing data. We used the same CNN as in MNIST with 10 as the output dimension.

**CIFAR-10**: CIFAR-10 (Krizhevsky & Hinton, 2009) is the labeled subsets of an 80 million tiny image dataset. This dataset consists of 60,000 32x32 color images in 10 classes, with 6,000 images per class. There are 50,000 training images and 10,000 test images. For CIFAR-10, we trained DenseNet (Huang et al., 2017) with a depth of 40 and a growth rate of 12.

For each dataset, the network was trained with a batch size of 64 for 100 epochs with a learning rate of 0.01 We used a weight decay of $1.0 \times 10^{-5}$ and the Nesterov optimization algorithm (Sutskever et al., 2013) with a momentum of 0.9. The network weights were initialized using the Glorot uniform (Glorot & Bengio, 2010).

### 3.3.2 RESULTS

Figure 4 shows the two-dimensional feature embeddings on the MNIST dataset. Different feature embeddings were acquired for each method. When softmax was used, the features spread in a fan shape and some part of the distribution overlapped around the origin. However, when the SDGM was used, the distribution for each class exhibited an ellipse shape and margins appeared between the class distributions. This is because the SDGM is based on a Gaussian mixture model and functions to push the samples into a Gaussian shape.

Table 2 shows the recognition error rates on each dataset. SDGM achieved better performance than softmax. As shown in Figure 4, SDGM can create margins between classes by pushing the features into a Gaussian shape. This phenomenon positively affected the classification capability.

## 4 RELATED WORK AND POSITION OF THIS STUDY

Figure 5 illustrates the relationship of our study with other studies. This study is primarily consists of three factors: discriminative model, Gaussian mixture model, and Sparse Bayesian learning. This study is the first that combines these three factors and expands the body of knowledge in these fields.

From the perspective of the sparse Bayesian classifier, the RVM (Tipping, 2001) is the most important related study. An RVM is combines logistic regression and sparse Bayesian learning. Since

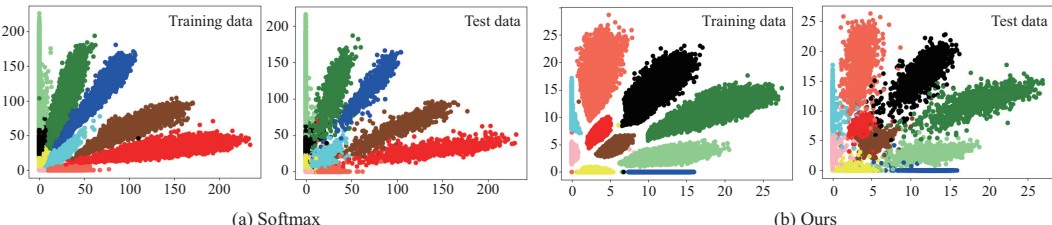

Figure 4: Visualization of CNN features on MNIST after end-to-end learning. In this visualization, five convolutional layers with four max pooling layers between them and a fully connected layer with a two-dimensional output are used. (a) results when a fully connected layer with the softmax function is used as the last layer. (b) when SDGM is used as the last layer instead. The colors red, blue, yellow, pink, green, tomato, saddlebrown, lightgreen, cyan, and black represent classes from 0 to 9, respectively. Note that the ranges of the axis are different between (a) and (b).

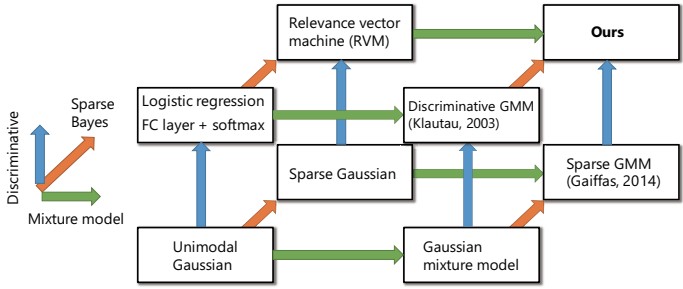

Figure 5: Relationship of our study with other studies.

the logistic regression is equivalent to the discriminative model of a unimodal Gaussian model, the SDGM can be considered as an extended RVM using a GMM. furthermore, from the perspective of the probabilistic model, the SDGM is considered as the an extended discriminative GMM (Klautau et al., 2003) using sparse Bayesian learning, and an extended sparse GMM (Gaiffas & Michel, 2014) using the discriminative model.

Sparse methods have often been used in machine learning. Three primary merits of using sparse learning are as follows: improvements in generalization capability, memory reduction, and interpretability. Several attempts have been conducted to adapt sparse learning to deep NNs. Graham (2014) proposed a spatially-sparse convolutional neural network. Liu et al. (2015) proposed a sparse convolution neural network. Additionally, sparse Bayesian learning has been applied in many fields. For example, an application to EEG classification has been reported (Zhang et al., 2017).

## 5 CONCLUSION

In this paper, we proposed a sparse classifier based on a GMM, which is named SDGM. In the SDGM, a GMM-based discriminative model was trained by sparse Bayesian learning. This learning algorithm improved the generalization capability by obtaining a sparse solution and automatically determined the number of components by removing redundant components. The SDGM could be embedded into NNs such as convolutional NNs and could be trained in an end-to-end manner.

In the experiments, we demonstrated that the SDGM could reduce the amount of weights via sparse Bayesian learning, thereby improving its generalization capability. The comparison using benchmark datasets suggested that SDGM outperforms the conventional sparse classifiers. We also demonstrated that SDGM outperformed the fully connected layer with the softmax function when it was used as the last layer of a deep NN.

One of the limitations of this study is that sparse Bayesian learning was applied only when the SDGM was trained stand-alone. In future work, we will develop a sparse learning algorithm for a whole deep NN structure including the feature extraction part. This will improve the ability of the CNN for larger data classification.

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

## A  APPENDIX

We explain that a fully connected layer with the softmax function, or logistic regression, can be regarded as a discriminative model based on a Gaussian distribution by utilizing transformation of the equations. Let us consider a case in which the class-conditional probability $P(\boldsymbol{x}|c)$ is a Gaussian distribution. In this case, we can omit $m$ from the equations (4)–(7).

If all classes share the same covariance matrix and the mixture weight $\pi_{cm}$, the terms $\pi_{cm}$ in (2), $x_1^2, x_1 x_2, \cdots, x_1 x_D, x_2^2, x_2 x_3, \cdots, x_2 x_D, \cdots, x_D^2$ in (3), and $-\frac{1}{2} s_{c11}, \cdots, -\frac{1}{2} s_{cDD}$ in (7) can be canceled; hence the calculation of the posterior probability $P(c|\boldsymbol{x})$ is also simplified as

$$P(c|\boldsymbol{x}) = \frac{\exp(\boldsymbol{w}_c^{\mathrm{T}} \boldsymbol{\phi})}{\sum_{c=1}^{C} \exp(\boldsymbol{w}_c^{\mathrm{T}} \boldsymbol{\phi})},$$

where

$$\boldsymbol{w}_c = [\log P(c) - \frac{1}{2} \sum_{i=1}^{D} \sum_{j=1}^{D} s_{cij} \mu_{ci} \mu_{cj} + \frac{D}{2} \log 2\pi + \frac{1}{2} \log |\boldsymbol{\Sigma_c}|,$$

$$\sum_{i=1}^{D} s_{ci1} \mu_{ci}, \cdots, \sum_{i=1}^{D} s_{ciD} \mu_{ci}]^{\mathrm{T}},$$

$$\boldsymbol{\phi} = [1, \boldsymbol{x}^{\mathrm{T}}]^{\mathrm{T}}.$$

This is equivalent to a fully connected layer with the softmax function.

