# OpenReview forum: "SDGM: Sparse Bayesian Classifier Based on a Discriminative Gaussian Mixture Model"
_ICLR.cc/2020/Conference — Reject_

### Official Review · AnonReviewer1 · 2019-10-07
**Official Blind Review #1**

**Rating:** 3

**Review:**

The paper presents an alternative to densely connected shallow classifiers or the conventional penultimate layers (softmax) of conventional deep network classifiers. This is formulated as a Gaussian mixture model trained via gradient descent arguments.

The paper is interesting and constitutes a useful contribution. The Gaussian mixture model formulation allows for inducing sparsity, thus (potentially) considerably reducing the trainable model (layer) parameters. The model is computationally scalable, a property which renders it amenable to real-world applications.

However, it is unfortunate that the paper does not take into account recent related advances in the field, e.g.

https://icml.cc/Conferences/2019/ScheduleMultitrack?event=4566

The paper should make this sort of related work review, discuss the differences from it, and perform extensive experimental comparisons.

**Experience Assessment:**

I have published in this field for several years.

**Review Assessment: Checking Correctness Of Derivations And Theory:**

I carefully checked the derivations and theory.

**Review Assessment: Checking Correctness Of Experiments:**

I carefully checked the experiments.

**Review Assessment: Thoroughness In Paper Reading:**

I read the paper thoroughly.

---

> ### Author Response · Authors · 2019-11-15
> **Response to Reviewer #1**
>
> Thank you so much for your constructive and positive comments. As pointed out, the relationship between our method and nonparametric Bayesian deep learning is very interesting. Unfortunately, due to the short rebuttal period, we could not include comparisons in the revised manuscript. We will investigate this in future work.

---

### Official Review · AnonReviewer3 · 2019-10-17
**Official Blind Review #3**

**Rating:** 3

**Review:**

This paper proposes a classifier, called SDGM, based on discriminative Gaussian mixture and its sparse parameter estimation.
The method aims to have a flexible decision boundary due to the mixture model as well as to generalize well owing to the sparse learning strategy.
While the method incorporates distinct ideas in the literature such as increased model complexity and use of a sparse prior, the paper needs more clearer explanation of the method design and careful empirical investigation.

My major concerns are as follows.
1) Explanation of the learning algorithm of Section 2.2 is unsatisfactory.
Specifically, the objective function and the principle of algorithm are unclear.
My guess is:
* c, m and T, z are equivalent. (T, z are one-hot representation of c, m.)
* The objective is to maximize the conditional probability of equation (1) given labeled data (x, t).
* This maximization is carried out in a similar way to the EM algorithm where m (or z) is regarded as a latent variable. This gives equation (9).
* A MAP estimate of parameter w is calculated using the prior (8).
Assuming the points above, I am still unsure about the following points:
1-a) How is the sparsity induced?
The prior of w seems to have the l2 regularizer, but how are \pi and r pulled toward zero?
1-b) How is the update of \alpha (17) derived?
Does this strengthen the sparsity by increasing \alpha given small \hat{w}?
What is the "orthogonal component" of \Lambda?
1-c) Is structure in w ignored?
Equation (7) indicates that parameter w has a certain structure such as nonnegative s_{cmii} or the determinant |\Sigma_{cm}| interacting with s_{cmij}.
In other words, the degree of freedom in w assuming the Gaussian likelihood of x is smaller than H.
These structure would be violated if the gradient descent or Newton's method is applied.
Do you mean by *discriminative* that we can freely set the parameter w?
Then, this point should be emphasized.

2) Which parameter is learned when combined into NN?
Parameters w and \pi?
How are these parameters made sparse in the end-to-end learning?

3) Ablation test to investigate which aspect impacts the performance.
Section 4 describes SDGM incorporates disciminative model, mixture model, and sparse Bayesian parameter estimation.
It is more informative to provide empirical results to see the impact of each property by comparing SDGM with, for example, RVM, discriminative GMM and sparse GMM.

Some other commets follow.
* \sqrt{\alpha_{cmh}} may be missing from the numerator of (8).
* For what distribution the expectation with regard to z is taken in e.g. (9, 11, 12, 13, 16)?
* What is D for CIFAR-10 experiment? DenseNet seems to use D=1000 units for the fully connected layer. Did the authors adopt this value?



**Experience Assessment:**

I have published one or two papers in this area.

**Review Assessment: Checking Correctness Of Derivations And Theory:**

I assessed the sensibility of the derivations and theory.

**Review Assessment: Checking Correctness Of Experiments:**

I carefully checked the experiments.

**Review Assessment: Thoroughness In Paper Reading:**

I read the paper at least twice and used my best judgement in assessing the paper.

---

> ### Author Response · Authors · 2019-11-15
> **Response to Reviewer #3**
>
> We genuinely thank the reviewer for the thorough, detailed, and insightful review. Below is our response to the question raised in the review:
> ---
>
> >>> Q: 1-a) How is the sparsity induced? The prior of w seems to have the l2 regularizer, but how are \pi and r pulled toward zero?
>
> A: Maximizing the denominator of Eq. (10), P(T, z| X, \alpha), induces the sparsity. Intuitively, this term represents the correlation between the teacher vector T and the hyperparameter \alpha. By maximizing this term, \alpha that is not correlated with T goes to infinite, thereby removing the corresponding w. The theoretical background of this technique is detained in [1]. Removing w of a certain component {c, m} decreases the value of P(c, m | x). This pulls \pi_ {cm} and r_{ncm} to zero because r_{ncm} = P(c, m | x_n)/ P(c, | x_n) and \pi_{cm} is the average of r_{ncm} over n (see Eq. (18)).
>
> [1] Michael E Tipping. Sparse Bayesian learning and the relevance vector machine. Journal of Machine
> Learning research, 2001.
> ---
>
> >>> Q: 1-b) How is the update of \alpha (17) derived? Does this strengthen the sparsity by increasing \alpha given small \hat{w}? What is the "orthogonal component" of \Lambda?
>
> A: Eq. (17) is derived by calculating the derivative of the normalization term of the Gaussian distribution that is derived via the Laplace approximation. We are so sorry that "orthogonal component" is a typo of “diagonal component.” We fixed it. Thank you for pointing it out.
> ---
>
> >>> Q: 1-c) Is structure in w ignored? Equation (7) indicates that parameter w has a certain structure such as nonnegative s_{cmii} or the determinant |\Sigma_{cm}| interacting with s_{cmij}. In other words, the degree of freedom in w assuming the Gaussian likelihood of x is smaller than H. These structure would be violated if the gradient descent or Newton's method is applied. Do you mean by *discriminative* that we can freely set the parameter w? Then, this point should be emphasized.
>
> A: Yes, the structure in w can be ignored since SDGM is trained as a discriminative model. According to [2, 3], the difference between discriminative and generative models can be explained by the implicit constraint on the model parameters. For example, in the generative Gaussian model, the parameter \mu should correspond to the mean of the distribution. In contrast, in the discriminative model, such constraint is removed, thereby reducing the statistical bias.
> [2] Tom Minka, Discriminative models, not discriminative training, Technical Report MSR-TR-2005-144, 2005.
> [3] Julia A. Lasserre et al., Principled Hybrids of Generative and Discriminative Models, IEEE Computer Society Conference on Computer Vision and Pattern Recognition (CVPR), 2006.
> ---
>
> >>> Q: 2) Which parameter is learned when combined into NN? Parameters w and \pi? How are these parameters made sparse in the end-to-end learning?
>
> A: Yes, w and \pi are learned in the combined model. Unfortunately, Bayesian sparse learning was not applied to the combined model due to the huge computational cost. This fact is described as a limitation in the Conclusion section.
> ---
>
> >>> Q: * \sqrt{\alpha_{cmh}} may be missing from the numerator of (8).
> A: Thank you very much for pointing it out. We fixed it.
>
> >>> Q: * For what distribution the expectation with regard to z is taken in e.g. (9, 11, 12, 13, 16)?
> A: z follows categorical distribution.
> ---
>
> >>> Q: * What is D for CIFAR-10 experiment? DenseNet seems to use D=1000 units for the fully connected layer. Did the authors adopt this value?
> A: If you mean D is the dimension of the final output (= the number of classes), we set it to 10 by changing the shape of the last fully connected layer.

---

### Official Review · AnonReviewer2 · 2019-10-24
**Official Blind Review #2**

**Rating:** 1

**Review:**

The paper proposes a discriminative Gaussian mixture model with a sparsity prior over the decoding weight. They can automatically learn the number of components with the sparsity prior and learn Gaussian-structured feature space.

1. I think the model is just ARD prior over discriminative GMM which is not that novel. DGMM models have been for a while [1,2]. Adding ARD sparsity prior over the decoding weight is also a classic routine. It's also well known that ARD can do feature selection and removal.

[1] Discriminative gaussian mixture models for speaker verification
[2] Discriminative Gaussian mixture models: A comparison with kernel classifiers

2. I don't think differentiating between discriminative GMM and generative GMM would make such a big deal. DGMM is basically Gaussian mixtures existing for each class. Any skill applied to GMM can be applied to DGMM. There are many works for component number selection for GMM with non-parametric Bayesian methods. For example, Dirichlet Process Mixture Model can automatically learn the number of components without predefining.

3. Only comparing SDGM with LR, SVM and RVM is quite weak, not mentioning that the performance is not that dominatingly better. SDGM is GMM+LR. So SDGM should be better than LR if the data has structures. What SVM you compare with? Do you use nonlinear kernels which can learn better nonlinear feature space?

Overall, I think the contribution of the paper is a bit incremental. I vote for a rejection.



**Experience Assessment:**

I have published in this field for several years.

**Review Assessment: Checking Correctness Of Derivations And Theory:**

I assessed the sensibility of the derivations and theory.

**Review Assessment: Checking Correctness Of Experiments:**

I assessed the sensibility of the experiments.

**Review Assessment: Thoroughness In Paper Reading:**

I read the paper thoroughly.

---

> ### Author Response · Authors · 2019-11-15
> **Response to Reviewer #2**
>
> We would like to thank the reviewer for going through the paper carefully and providing useful feedback to our work. The followings are our responses to the concerns raised in the review:
> ---
> >>> Q: I think the model is just ARD prior over discriminative GMM which is not that novel. DGMM models have been for a while [1,2]. Adding ARD sparsity prior over the decoding weight is also a classic routine. It's also well known that ARD can do feature selection and removal.
>
> A: As pointed out, the proposed SDGM can be considered as a combination of ARD prior and DGMM. Whereas many studies have investigated to combine ARD prior with a discriminative model, the combination of ARD prior and a discriminative *mixture* model is not trivial. The difficulty of fusing them is that we should solve the maximization of posterior probability, optimization of ARD prior, and component assignment of each sample, simultaneously. This cannot be solved if we just incorporate ARD prior into DGMM. To solve this problem, we developed a new learning algorithm by nesting ARD prior updating into the EM algorithm, which is shown in Algorithm 1.
>
> ---
> >>> Q: I don't think differentiating between discriminative GMM and generative GMM would make such a big deal. DGMM is basically Gaussian mixtures existing for each class. Any skill applied to GMM can be applied to DGMM. There are many works for component number selection for GMM with non-parametric Bayesian methods. For example, Dirichlet Process Mixture Model can automatically learn the number of components without predefining.
>
> A: The difference between discriminative and generative models is clearly formulated in the literature [1, 2]. According to the literature, the difference between discriminative and generative models can be explained by the implicit constraint on the model parameters. For example, in the generative Gaussian model, the parameter \mu should correspond to the mean of the distribution. In contrast, in the discriminative model, such constraint is removed, thereby reducing the statistical bias. Therefore, the discriminative and generative models should be considered as different models.
> [1] Tom Minka, Discriminative models, not discriminative training, Technical Report MSR-TR-2005-144, 2005.
> [2] Julia A. Lasserre et al., Principled Hybrids of Generative and Discriminative Models, IEEE Computer Society Conference on Computer Vision and Pattern Recognition (CVPR), 2006.
>
> ---
> >>>Q: Only comparing SDGM with LR, SVM and RVM is quite weak, not mentioning that the performance is not that dominatingly better. SDGM is GMM+LR. So SDGM should be better than LR if the data has structures. What SVM you compare with? Do you use nonlinear kernels which can learn better nonlinear feature space?
>
> A: As we explained in Section 4, SDGM can be considered as the mixture version of RVM. In the original paper of RVM (Tipping 2001), the author proposed RVM as the Bayesian version of SVM and showed that RVM can obtain a sparser solution than SVM. Based on this background, we employed SVM and RVM to compare the sparsity of the trained models.

---

### Decision · Program_Chairs · 2019-12-19

**Decision:**

Reject

**Comment:**

This paper presents a method for merging a discriminative GMM with an ARD sparsity-promoting prior.  This is accomplished by nesting the ARD prior update within a larger EM-based routine for handling the GMM, allowing the model to automatically remove redundant components and improve generalization.  The resulting algorithm was deployed on standard benchmark data sets and compared against existing baselines such as logistic regression, RVMs, and SVMs.

Overall, one potential weakness of this paper, which is admittedly somewhat subjective, is that the exhibited novelty of the proposed approach is modest.  Indeed ARD approaches are now widely used in various capacities, and even if some hurdles must be overcome to implement the specific marriage with a discriminative GMM as reported here, at least one reviewer did not feel that this was sufficient to warrant publication.  Other concerns related to the experiments and comparison with existing work.  For example, one reviewer mentioned comparisons with Panousis et al., "Nonparametric Bayesian Deep Networks with Local Competition," ICML 2019 and requested a discussion of differences.  However, the rebuttal merely deferred this consideration to future work and provided no feedback regarding similarities or differences.  In the end, all reviewers recommended rejecting this paper and I did not find any sufficient reason to overrule this consensus.